# Machine-learning reprogrammable metasurface imager

Lianlin Li [1], Hengxin Ruan[1], Che Liu [2], Ying Li[3], Ya Shuang[1], Andrea Alù [4,5,6], Cheng-Wei Qiu[3] & Tie Jun Cui[2]

Conventional microwave imagers usually require either time-consuming data acquisition, or complicated reconstruction algorithms for data post-processing, making them largely ineffective for complex in-situ sensing and monitoring. Here, we experimentally report a real-time digital-metasurface imager that can be trained in-situ to generate the radiation patterns required by machine-learning optimized measurement modes. This imager is electronically reprogrammed in real time to access the optimized solution for an entire data set, realizing storage and transfer of full-resolution raw data in dynamically varying scenes. High-accuracy image coding and recognition are demonstrated in situ for various image sets, including hand-written digits and through-wall body gestures, using a single physical hardware imager, reprogrammed in real time. Our electronically controlled metasurface imager opens new venues for intelligent surveillance, fast data acquisition and processing, imaging at various frequencies, and beyond.

---

[1] State Key Laboratory of Advanced Optical Communication Systems and Networks, Department of Electronics, Peking University, 100871 Beijing, China. [2] State Key Laboratory of Millimeter Waves, Southeast University, 210096 Nanjing, China. [3] Department of Electrical and Computer Engineering, National University of Singapore, 4 Engineering Drive 3, Singapore 117583, Singapore. [4] Photonics Initiative, Advanced Science Research Center, City University of New York, 85 St. Nicholas Terrace, New York, NY 10031, USA. [5] Physics Program, The Graduate Center, City University of New York, 365 Fifth Avenue, New York, NY 10016, USA. [6] Department of Electrical Engineering, City College of New York, New York, NY 10031, USA. These authors contributed equally: Lianlin Li, Hengxin Ruan, Che Liu, Ying Li. Correspondence and requests for materials should be addressed to L.L. (email: lianlin.li@pku.edu.cn) or to A.Aù. (email: aalu@gc.cuny.edu) or to C.-W.Q. (email: chengwei.qiu@nus.edu.sg) or to T.J.C. (email: tjcui@seu.edu.cn)

Efficient microwave imaging systems are becoming increasingly important in modern society, however, rapid processing of information and data poses new challenges for current imaging techniques[1-7]. In many cases, such as for security screening or pipeline monitoring, the flux of data is so large that full-resolution imaging is inefficient and unmanageable, and represents a huge waste of resources and energy, since only a few properties of the images are actually of interest, such as the position of an object, or its dynamic changes[8-13]. These situations require an imaging device that can instantly reconstruct scenes in an intelligent and efficient way, i.e., rendering the important feature extraction with high speed, fidelity, and compression ratio, as it commonly happens in biological systems, such as our brain. In the past decade, a wide variety of computational imagers[1-10] have been introduced to perform image compression at the physical level, thereby eliminating the need for storage, transfer and processing of the full-pixel original scenes. In this case, just a few relevant data are recorded to reconstruct the scene without losing the information of interest, which is particularly useful for microwave or millimeter-wave radars.

However, microwave imagers to date always have to compromise between speed and image quality (fidelity and compression ratio)[1-7]. Recently, compressive sensing inspired computational imagers[1-5] have been proposed to reduce remarkably hardware cost and speed data acquisition, which are at the cost of iterative reconstruction algorithms. The optimization solves a time and resource consuming inverse problem for each individual scene. In this sense, it is typically required to solve the inverse scattering problem again and again when the scene changes, making them largely ineffective for complex in-situ sensing and monitoring.

To properly process a large data flux of high complexity, it is necessary to study and extract the common features across the data set. Such a challenge has been remarkably tackled in recent times by emerging techniques in machine learning[9-16]. Recent advances in optics show that the machine learning can be utilized to conduct the measurements such that the high-quality imaging and high-accuracy object recognition with a few measurements can be realized[9-13]. However, a gap exists between machine-learning techniques and their direct employment in physical-level microwave imaging, due to the restricted configurations of the imagers mentioned above. Specifically, almost all microwave imagers cannot produce the radiation patterns corresponding to the machine-learning optimized measurement modes in real-time and cost-efficient way. Moreover, these imagers are primarily designed to accomplish imaging functions for a specified theme, which cannot be altered after fabrication[1-3]. It would be of great benefit to develop a single imaging device that can achieve all machine-learning-desired radiation patterns and that can switch its functional theme simply by training with samples of a new target group. Inspired by recently introduced digital coding and programmable metamaterials and metasurfaces[17-20], we propose and experimentally realize a microwave reprogrammable digital metasurface that enables optimal imaging quality processing based on machine-learning, and reconfigure itself for a wide variety of scenes.

The machine-learning imaging metasurface heavily relies on the opportunities opened by the flexible and dynamic manipulations of EM waves in real time. Metasurfaces, ultrathin planar devices composed of subwavelength metallic or dielectric meta-atoms (resonators), have shown remarkable opportunities to arbitrarily tailor the propagation and scattering of EM wavefronts[21-30]. Metasurfaces have been recently proposed to realize versatile functions, including spin-Hall effect on light[21], ultrathin planar lenses[22-24], metasurface-based holograms[25,26], imaging[1-5,27], and optical vortex beam generators[28,29]. Most recently, reprogrammable digital coding metamaterials have been proposed to dynamic holography[18], compressive imaging[19], beam scan[20], and other related operations[20], in which the coding sequence of the meta-atoms with digitized states have been used to manipulate the impinging microwave waves in an efficient and controllable fashion.

In this article, we propose microwave reprogrammable imager by combing machine learning techniques with 2-bit coding metasurface, which are confirmed by proof-of-concept experiments. The proposed microwave imager, after a period of training, can produce high-quality imaging and high-accuracy object recognition operating on the compressed measurements that are obtained directly from the imager, without the need for costly computationally image reconstruction. Our real-time imaging will pave the way for compressed imaging applications in the microwave, millimeter-wave, and terahertz frequencies, and beyond.

## Results

**Principle of machine-learning imager.** Typically, microwave imaging can be used to recognize a scene from measurements of the scattered fields. Solving this inverse problem requires to establish a model bridging the measured return signal and the scene[1], which is typically expressed as $\mathbf{y} = \mathbf{Hx} + \mathbf{n}$, where $\mathbf{y} \in \mathbb{C}^M$ denotes the measurements, $\mathbf{H}$ characterizes the measurement (or sensing) matrix, $\mathbf{x} \in \mathbb{C}^N$ denotes the scene under investigation, and $\mathbf{n}$ is the measurement noise. Each row of $\mathbf{H}$ corresponds to one measurement mode, and the number of rows equals the number of measurements. This problem can be viewed as linear embedding in machine learning[14-16], as discussed in Methods. In light of the established theory of machine learning, the measurement modes may be efficiently learned from the training samples relevant to the scene of interest, if they are available, such that they can provide as much information of the scene as possible, see Fig. 1a. In this way, well-trained measurement modes are responsible for producing high-quality images or/and high-accuracy classification from a remarkably reduced number of measurements, i.e., $M \ll N$.

**Design of real-time digital-metasurface imager.** The proposed reprogrammable imager heavily relies on a dynamic wavefield spatial modulator. In order to manipulate arbitrarily the measurement modes needed by machine learning with high accuracy in real time, we propose an electrically modulated 2-bit reprogrammable coding metasurface. In order to illustrate the operational principle, the 2-bit coding metasurface is designed to work at the frequency around 3 GHz. We designed and fabricated a reprogrammable metasurface composed of a two-dimensional array of electrically controllable meta-atoms (see Supplementary Figs. 1, 2). Each macro meta-atom can be independently tuned, and an incident illumination on metasurface with different coding sequences will result in different backward scattered fields. Supplementary Fig. 2d depicts the designed meta-atom, with a size of $15 \times 15 \times 5.2$ mm, which is composed of three subwavelength-scale square metallic patches printed on a dielectric substrate (Rogers 3010) with dielectric constant of 10.2. Any two adjacent patches are connected via a PIN diode (BAR 65–02L), and each PIN diode has two operation states controlled by the applied bias voltage. Illuminating by an $x$-polarized plane wave, such a meta-atom supports four different phase responses, denoted as "00" (state 0), "01" (state 1), "10" (state 2), "11" (state 3), and determined by controlling the ON/OFF states of the three PIN diodes in a suitable combination, corresponding to four digitized phase levels 0, $\pi/2$, $\pi$, and $3\pi/2$. The diode is at the state ON (or OFF) with an applied external bias voltage of 3.3V (or 0V). To isolate the DC feeding port and microwave signal, three

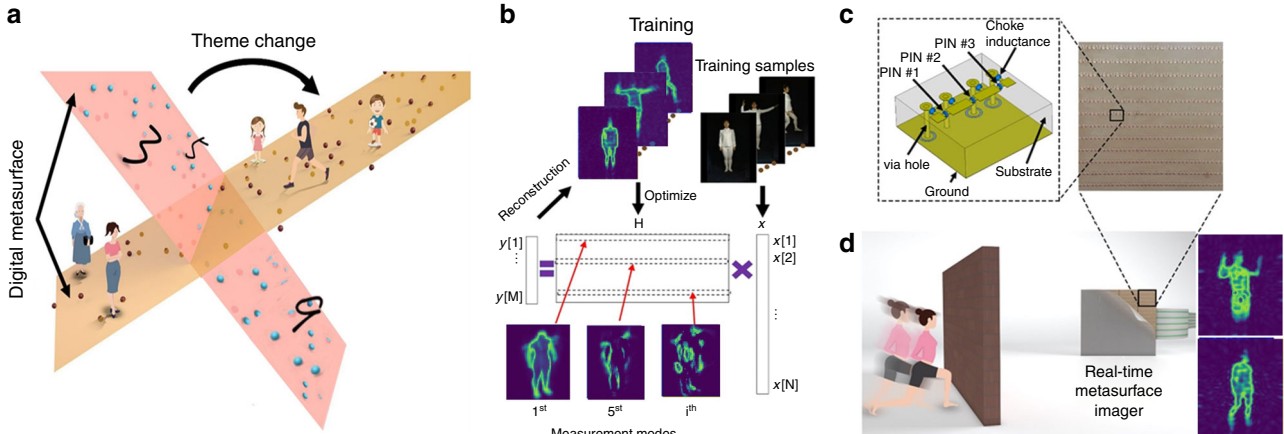

**Fig. 1** Working principle of the real-time digital-metasurface imager. **a** The proposed machine-learning metasurface imager can be optimized for different kinds of scenes. The optimization is performed by training the manifold representing the metasurface configuration (the surfaces) to be close to the training data (scatters). **b** The illustration of training the reprogrammable imager by using the PCA method (see Supplementary Note 2), in which the training person is Hengxin Ruan (coauthor). **c** The map of coding metasurface and the unit cell (see Supplementary Fig. 1, 2). **d** The illustration of real-time reprogrammable metasurface imager imaging a moving person behind a wall. Two reconstructions are displayed at right (more experimental results in Supplementary Videos 3, 4 and 5)

choke inductors of 30nH are used in each meta-atom. When the PIN diode is in the ON state, it is equivalent to a series circuit with parasitic inductance and resistance; while in the OFF state it is equivalent to a series circuit with parasitic inductance, capacitance, and resistance.

Supplementary Fig. 2e, f show the phase and amplitude responses as a function of frequency when the meta-atom works in four different states, as calculated by commercial software, CST Microwave Studio, Transient Simulation Package. It can be observed that the phase difference between two neighboring states falls in the range (90°−15°, 90°+15°) around the frequency of 3.2 GHz. As a consequence, these meta-atoms can be treated as unit cells of a 2-bit digital coding metasurface, mimicking "00", "01", "10" and "11". We further observe from Supplementary Fig. 2f that the simulated reflection can reach a high efficiency above 85% for all digital states around 3.2GHz.

The proposed reprogrammable imager is trainable in the sense that the reprogrammable coding metasurface can be controlled by the field programmable gate array (FPGA) such that the desirable radiation patterns (i.e., the measurement modes) are matched with that required by machine learning. To that end, we explored a straightforward two-step strategy: 1) the desirable radiation patterns are firstly trained by using the machine learning technique; 2) the corresponding coding patterns of metasurface are designed from the obtained radiation patterns. To design the 2-bit coding patterns generating the desirable radiation patterns, which is a NP-hard combinatorial optimization problem, we apply the modified Gerchberg-Saxton (GS) algorithm designed for the discrete-valued optimization problem[26]. Critically, the four-phase quantization of the metasurface is performed in each iteration of forward/backward propagation. In addition, an effective induced current is introduced to characterize the realistic response of each meta-atom to address the accuracy issue arising from the point-like dipole model, which can be obtained by applying the Huygens principle or the source inversion technique (see Methods). Machine-learning guided imaging for the first proof-of-concept demonstration, we use the developed imager to realize machine-learning-guided imaging. Throughout the article, two popular linear embedding techniques, i.e., the random projection[13,15] and the principal component analysis (PCA)[12], are applied to train our machine-learning imager. Apparently, it is trivial to conceive a random projection matrix by independently

and randomly setting the status of PIN diodes of the reprogrammable metasurface. However, for PCA measurements, the status of PIN diodes needs to be carefully manipulated to achieve the desired measurement modes.

We experimentally examine the performance of the developed machine-learning imager by monitoring the movement of a person in front of the metasurface. In our study, we use a moving person to train our machine-learning imager, and use another moving person to test it. The training person, armed with and without glass scissors, is recoded in Supplementary Videos 1 and 2, respectively. The whole training time is less than 20 min with the proposed proof-of-concept system; however, such time could be remarkably reduced with a specialized receiver, as discussed below. After being trained, the machine-learning imager can produce the measurement modes required by PCA. The 16 PCA leading measurement modes (i.e., the radiation patterns of the reprogrammable metasurface) and the corresponding metasurface coding patterns are reported in Fig. 2b, c, respectively. Additionally, the corresponding theoretical PCA bases are reported in Fig. 2a as well. Here, the radiation patterns, at the distance of 1 m away from the metasurface, are experimentally obtained by performing the near-field scanning technique in combination with the near-to-far field transform (see Methods). Figure 2a–c demonstrate that the machine-learning imager is capable of generating the measurement modes required by PCA, which establishes a solid foundation for the machine-learning-driven imaging with significantly reduced measurements.

Next, the trained machine-learning imager was used to monitor another moving person. The series of reconstructions of the test person with and without the glass scissor are recorded in Supplementary Videos 3 and 4, respectively. This set of videos shows that not only the gestures of the test person can be recovered by the machine-learning imager, but also the armed glass scissor can be clearly reconstructed. In order to show the "see-through-the-wall" ability of the proposed machine-learning imager, Supplementary Video 5 shows that the proposed machine-learning imager can clearly detect the continuous movement of the test person behind a 3cm-thickness paper wall. In all results, 400 PCA-measurements are used. Supplementary Video 6 provides the evolution of the images with the growth of measurements. This video demonstrates that the machine-learning imager, trained with PCA, can produce high-quality

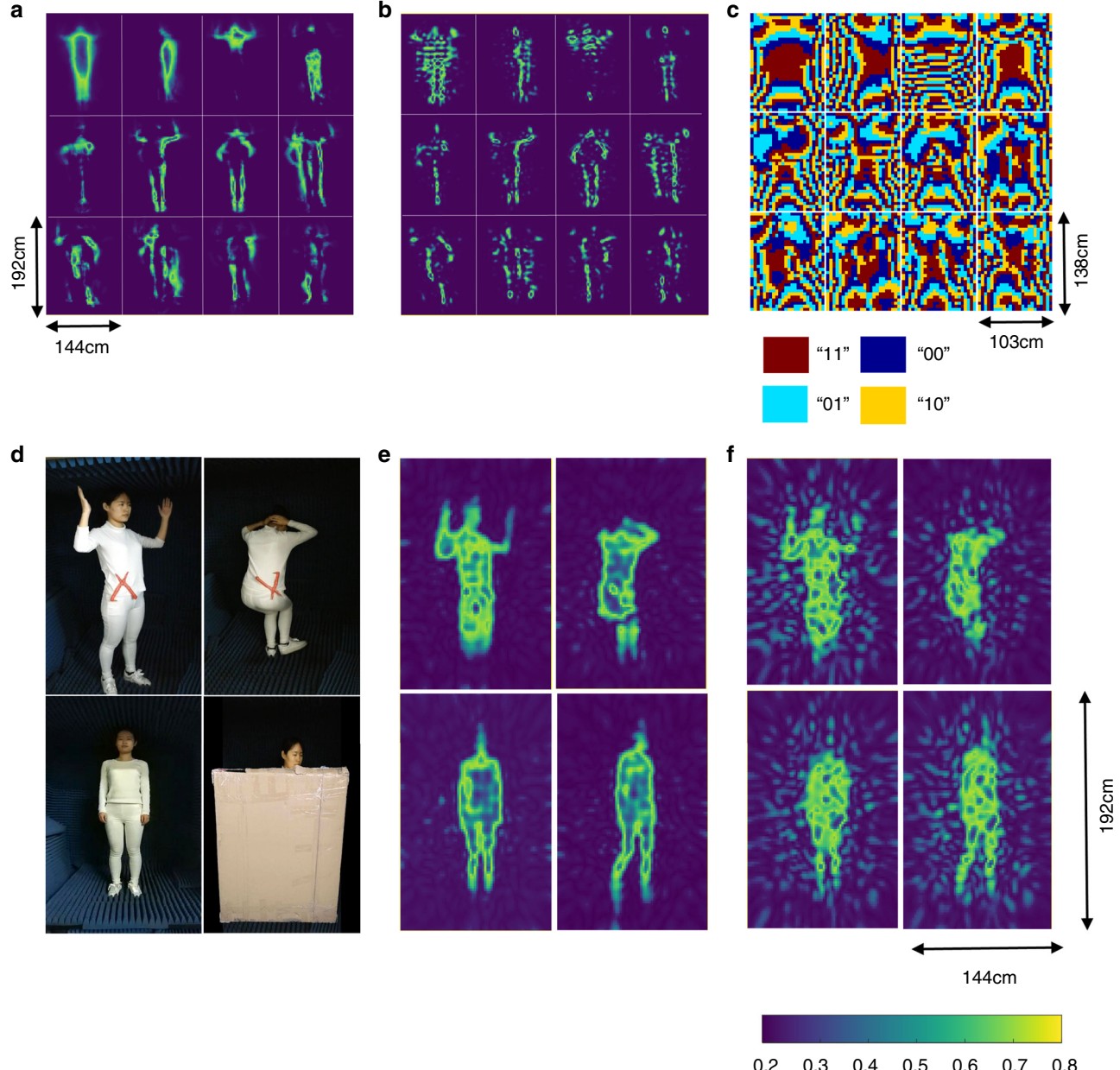

**Fig. 2** Configurations and results for human body imaging. (**a**) and (**b**) report 16 leading modes of theoretical PCA and the corresponding experimental patterns radiated by the machine-learning imager, respectively. **c** shows the coding patterns of 2-bit coding metasurface corresponding to **b**. **d** shows four images of a testing person (Ya Shuang, coauthor), where the top two images are armed with glass scissors, as shown in red. **e** and **f** are the reconstructed images corresponding to **d** using the machine-learning imagers trained with PCA and random projection, respectively, where 400 measurements are used. Note that the image results have been normalized by their own maximums

images with only 400 measurements, far fewer than the number of 8000 unknown pixels. Selected reconstructions from Supplementary Videos 3–6 are plotted in Figs. 2, 3. In order to show the benefit gained by the machine-learning metasurface imager trained with PCA over the random projection, Figs. 2, 3 provide corresponding results when the reprogrammable metasurface is encoded in a random manner, i.e., the measurement modes **H** behave as a random matrix. This set of figures clearly demonstrates that the reconstructions by PCA have overwhelming advantages over those by the random projection in cases of small amount of measurements, since a large amount of relevant training samples are incorporated in PCA. Note that both the head and feet cannot be clearly identified due to the limited FOV,

which probably arises from two reasons: 1) the use of a directive receiver (horn antenna); and 2) the limited size of the metasurface with respect to the person under consideration. Nonetheless, it can be faithfully deduced from the previous results that the machine-learning imager, after being well trained, can be used to implement the real-time and high-quality imaging from the considerably reduced measurements.

**Object recognition in compressed domain.** As a second proof-of-concept demonstration, we consider object recognition in the compressed domain using the machine-learning metasurface imager, which avoids the computational burden of costly image reconstructions. Here, a simple nearest-neighbor algorithm[11,12] is

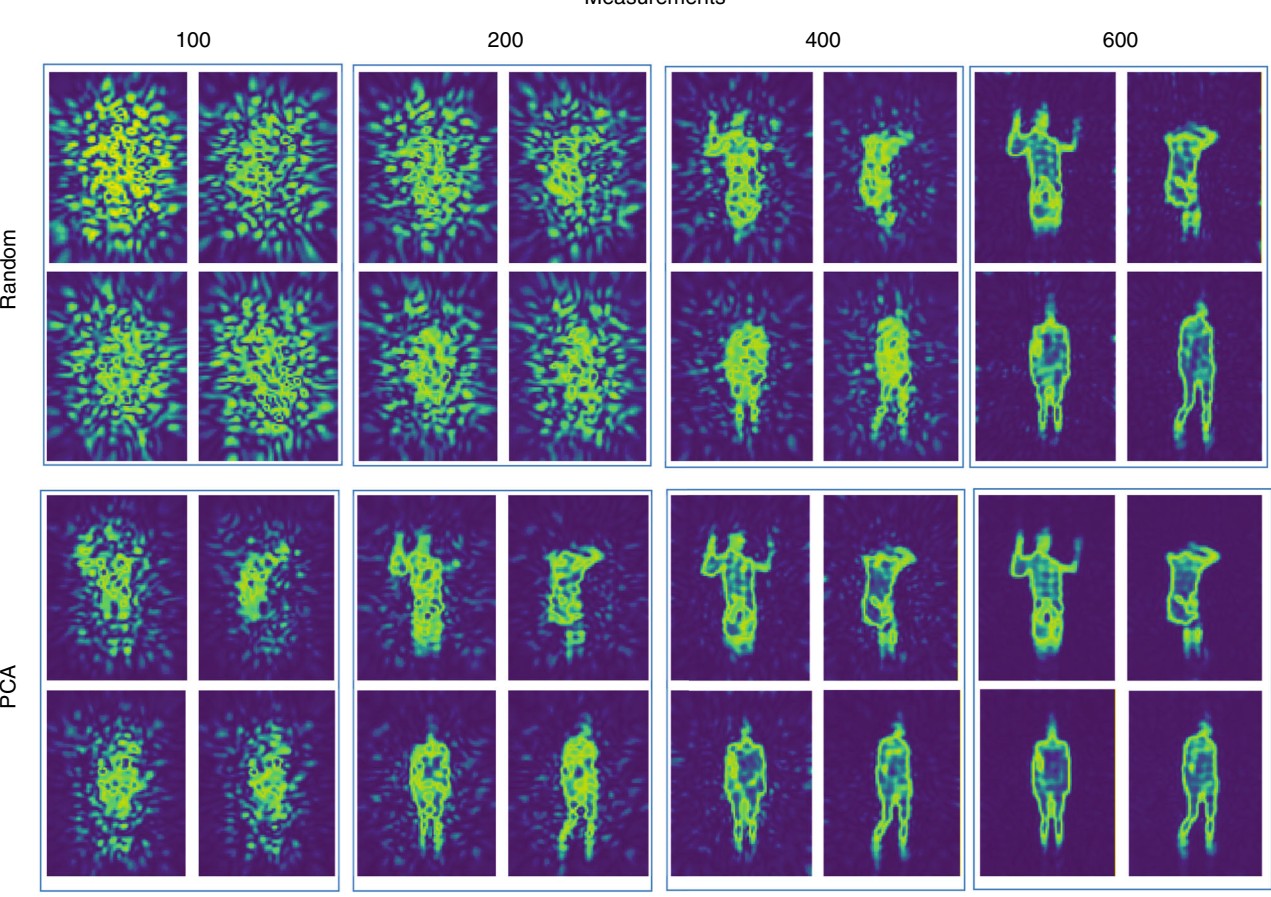

**Fig. 3** Machine-learning-driven imaging results of four different cases in Fig. 2d. Different numbers of measurements, 100, 200, 400, and 600, were performed with linear embedding techniques of random projection and PCA. More detailed results are recorded in Supplementary Videos 3 and 4. This set of panels clearly demonstrates that the reconstructions by PCA have overwhelming advantages over those obtained by the random projection for small number of measurements, since a large amount of relevant training samples are incorporated in PCA

used to recognize the input image. In contrast to the conventional machine learning techniques consisting of fully digital sampling followed by data post-processing, our technique for object recognition is directly implemented at the physical level by using our reprogrammable metasurface.

Assume that there are $K$ classes of labeled samples, and the center of the $k$-th class is $\tilde{\mathbf{y}}_k = \frac{1}{C_k}\sum_{t=1}^{C_k}\mathbf{Hx}_k^t, k = 1, 2, \ldots, K$, where $C_k$ is the total number of samples in the $k$-th class. For a given image $\boldsymbol{x}$ and its reduced measurements $\mathbf{y} \approx \mathbf{Hx}$, the object recognition consists of comparing $\boldsymbol{y}$ with the stored vectors $\tilde{\mathbf{y}}_k(k = 1, 2, \ldots, K)$ and selecting the one with the closest match, which is classified mathematically with the minimum Euclidean distance $c = \operatorname{argmin}_k \left( \|\mathbf{y} - \tilde{\mathbf{y}}_k\|_2^2 \right)$[11,12]. This concept is implemented in the proposed machine-learning metasurface imager, from which a real-time object recognition system is developed. Once again, the two aforementioned machine-learning techniques (random projection and PCA) are applied to train the proposed machine-learning imager.

Figure 4a investigates the classification performance of the machine-learning imager versus the number of measurements, as the imager is trained by the random projection (green dashed line) and PCA (red line with plus). For comparison, the theoretical PCA (blue straight line) result is also provided. Here, the classification rate is calculated by taking the average over three human-body gestures (standing, bending and raising arms),

in which each human-body gesture has over 1000 test samples. Fig. 4b compares specific classification results for the random projection, PCA, and theoretical PCA, where 25 measurements are used. Note that each period of data acquisition is about 0.016 s, and thus the whole data acquisition of 25 measurements is around 0.4 s. The classification performance of PCA is much better than the random projection, and it quickly approaches to the ideal result with more measurements. In particular, 25 measurements are enough to get objection recognition of the three human body actions, when the machine-learning imager is trained with PCA. This set of figures clearly indicates that the machine-learning imager trained by PCA can achieve the theoretical upper limit of classification. Moreover, when the machine-learning imager is trained by PCA, acceptable classification results can be obtained with around 60 measurements, corresponding to a compression rate of only 7.6%.

## Discussion

In this article, we have presented the concept and realization of a reprogrammable imager based on a 2-bit reprogrammable coding metasurface. We have shown that our device, when trained with machine-learning techniques at the physical level, can be utilized to monitor and recognize the movement of human in a real-time manner with remarkably reduced measurements. In order to validate the performance of our imager, we have experimentally tested our machine-learning imager with a benchmark dataset, the MNIST dataset, widely used in the community of machine

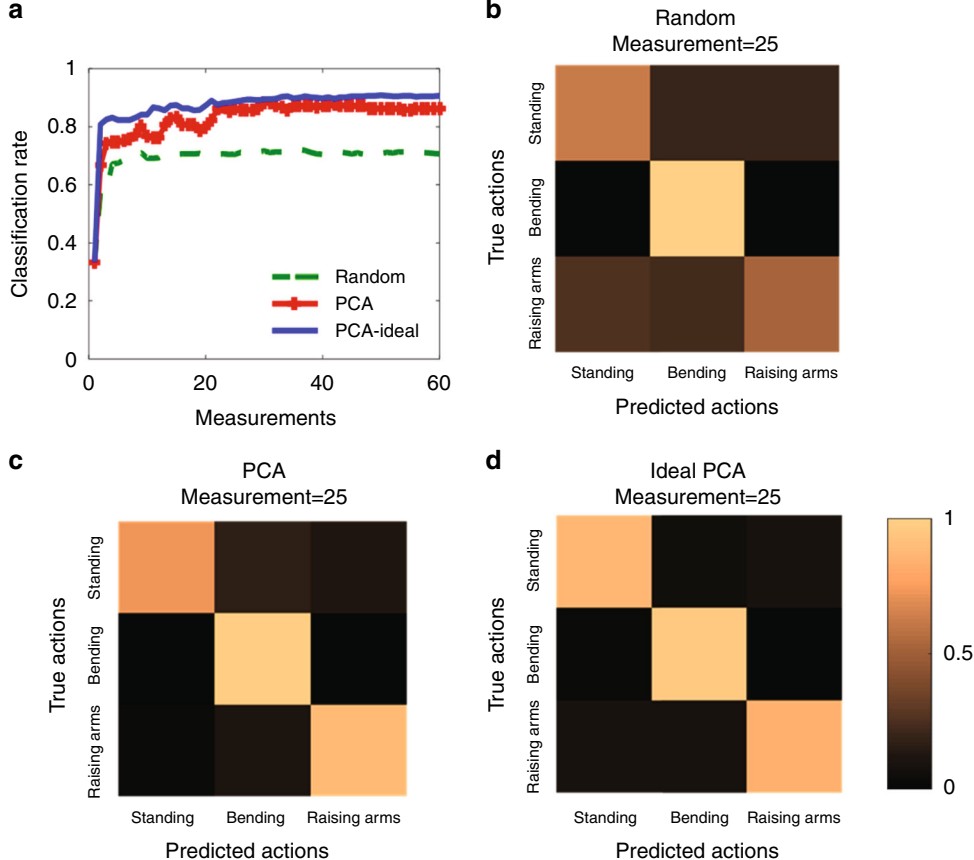

**Fig. 4** Classifying body gestures with the proposed imager. **a** Classification rate versus number of measurements when the programmable metasurface is trained by random projection and PCA. For comparison, the corresponding results by theoretical PCA (blue line) are also provided. **b**–**d** The comparison of specific classification results by (**b**) random projection, (**c**) PCA, and (**d**) theoretical PCA, in which 25 measurements are used

learning[14–16], which consists of ten handwritten digits from 0 to 9. We fabricated 5000 digit-like PEC objects according to the profiles of digits in the MNIST dataset, which has $28 \times 28$ pixels, with resolution of $3 \times 3$cm. As in the previous examples, we consider the linear embedding techniques of PCA and random projection. The theoretical PCA measurement modes and the corresponding experimental radiation patterns of our machine-learning imager are shown in Supplementary Fig. 3a, b respectively. Supplementary Fig. 3f illustrates the recovered images for ten digit-like objects by the proposed machine-learning imager trained with PCA and random projection with varying numbers of measurements. To quantitatively evaluate the reconstruction with different training mechanisms (PCA or random projection), signal-to-noise-ratios (SNRs)[31] of reconstructions as functions of measurements are plotted in Supplementary Fig. 3g–i. We clearly observe that the quality of image turns better with the increasing measurements, and PCA behaves better than the random projection, especially in low measurement cases. Moreover, when the machine-learning imager is trained by PCA, the acceptable results can be achieved with just 100 measurements, corresponding to a compression rate of 12.8%. Here, the compression rate is defined as the number of measurements over that of imaging pixels.

Supplementary Fig. 4a reports the average classification rate of the ten digit-like objects with varying measurements, in which the theoretical classification results of PCA operating on the ground truths are provided, referred to theoretical PCA. The experimental classification results for the ten digits for two aforementioned machine learning techniques are detailed in Supplementary Fig. 4b, c with 60 measurements, respectively.

This set of figures clearly indicates that the machine-learning imager trained by PCA can achieve the theoretical upper limits of classification. In addition, they are remarkably higher than that of the random projection and, when the machine-learning imager is trained by PCA, acceptable classification results can be obtained with around 60 measurements, corresponding to a compression rate of 7.6%.

In summary, a 2-bit programmable coding metasurface was proposed and experimentally realized, which can generate the radiation patterns required by machine learning techniques to realize a groundbreaking imaging hardware platform. A prototype of the machine-learning imager was fabricated to realize the real-time machine-learning-guided imaging and compressed-domain object recognition. The proof-of-concept experimental results agree very well with the numerical simulations, validating the programmable metasurface as a viable means for creating sophisticated images and high-accuracy object classification with significantly reduced measurements. In our experiments, the switching speed of the PIN diode is 3 ns, and the FPGA clock rate is 300 MHz. We use a vector-network analyzer (VNA) for data acquisition, and the whole data acquisition time is almost entirely taken by the VNA itself, i.e., around 0.016 s each operation cycle. If a specialized receiver were employed, the data acquisition time would be significantly reduced. For instance, taking the commercial continuous-wave radar technique as an example, the sampling frequency is usually about 1MHz, corresponding to about 0.1 milliseconds for sampling 100 data points. In this way, the data acquisition can be significantly speeded up by a factor of 100 and beyond.

Dynamic/active metamaterials[30–36] using tunable or switchable materials have been proposed to realize various functionalities in the terahertz frequency[30,32], near-infrared[33,34], and beyond[31,35,36], such as thermal-sensitive phase-change materials (i.e., $Ge_2Sb_2Te_5$)[31], gated graphene[30], and mechanical actuation[35]. Hence, in combination with switchable materials[30–36], we envision that our approach can be extended to millimeter-wave, terahertz frequency, and beyond to realize the next generation of efficient imagers.

## Methods

**Design of the 2-bit metasurface unit.** Supplementary Fig. 1 shows a sketch of the designed 2-bit digital coding meta-atom, and more detailed parameters are provided in Supplementary Fig. 2. The 2-bit digital coding metasurface is designed to operate around 3 GHz. Three PIN diodes (BAR 65–02L) are loaded to control the reflection phase of metasurface, and the equivalent circuits of the diode for the ON and OFF states are reported in Supplementary Fig. 2. Finite-element simulations in commercial software (CST Microwave Studio) were used to design the meta-atom. The reflection response of the meta-atom was investigated under different operation states of the PIN diodes. A Floquet port is used to produce an x-polarized wave incident on the metasurface and monitor the reflected wave. Periodic boundary conditions are set to the four sides to model an infinite array. $3 \times 3$ units with the same state are grouped into a macro meta-atom so that mutual coupling can be reduced.

**Inverse problem model.** As shown in Supplementary Note 1, the radiated electrical field **E** is linearly related to the current **J** induced over the metasurface by plane waves[26]. The source inversion method is employed to calibrate the current distributions of $\mathbf{J}^{(00)}$, $\mathbf{J}^{(01)}$, $\mathbf{J}^{(10)}$, and $\mathbf{J}^{(11)}$. Then the metasurface particle either at the state of '00', or '01', or '10', or '11' is illuminated by the plane wave, and the resulting co-polarized electric field is collected at one wavelength away from the metasurface particle. The calibration measurements are organized into a column vector **e**. To characterize the inhomogeneous current induced on metasurface particle, the particle is regarded as an array of $3 \times 3$ point-like dipoles, stacked in a 9-length column vector of **p**. Apparently, the calibration vector **e** is linearly related to the vector **p** through $\mathbf{e} = \mathbf{Gp}$, where the entries of **G** come from the three-dimensional Green's function in free space. Hence the complex-valued amplitude of **p** can be readily retrieved by solving the least-square problem of $\mathbf{e} = \mathbf{Gp}$.

**Linear machine learning techniques.** From the perspective of machine learning, the reduced measurements can be regarded as linear embedding of the probed scene in the low-dimensional space. Given a sample $\mathbf{x} \in \mathcal{X}$, where $\mathcal{X}$ is a cloud of Q points in an N-dimensional Euclidean space $\mathbb{R}^N$, the low-dimensional linear embedding method consists of finding a projection matrix **H** such that the M-dimensional projection $\mathbf{y} = \mathbf{Hx}$ has as small loss of intrinsic information as possible compared to **x**, where $M \ll N$. The random projection has attracted intensive attention in the past decade, in which the entries of **H** are drawn from independent random numbers. Our random approach imposes no restrictions on the nature of the object to be reconstructed. On the contrary, PCA is a prior data-aware linear embedding technique, where each row of **H** (i.e., the measurement mode) is trained over many training samples available. In this way, when a set of prior knowledge on the scene under investigation are available, the PCA approach enables the design of efficient measurement matrices, allowing the number of measurements to be limited compared to a purely random system.

**Imaging modeling.** The Born approximation assumes that the field scattered from a point in the scene is simply proportional to the incident field at that point. Based on this approximation[1–7], the entry of the measurement matrix ($H_{ij}$) is simply proportional to the fields radiated by the transmitter and receiver antennas at a given point in the scene $\mathbf{r}_j$: $H_{ij} \propto \mathbf{E}_i^T(\mathbf{r}_j) \cdot \mathbf{E}_i^R(\mathbf{r}_j)$, where $\mathbf{E}_i^T(\mathbf{r}_j)$ and $\mathbf{E}_i^R(\mathbf{r}_j)$ represent the radiation and receiving patterns related to the coding metasurface, respectively. Each row of the measurement matrix corresponds to a measurement mode, and hence the number of rows equals the number of measurements, and the number of columns equals the number of voxels in the scene to be reconstructed. Specifically, the number of measurement modes is determined by the coding patterns of the reprogrammable digital metasurface. The imaging solution reads $\hat{\mathbf{x}} = \mathbf{H}^+\mathbf{y}$, where $\mathbf{H}^+$ is the pseudo-inverse matrix of **H**.

**Measurement system.** This subsection discusses the system configuration of the proposed programmable imager. The schematic overview is given in Supplementary Fig. 1, and is detailed below. The system configuration consists of four main blocks: host computer, FPGA microcontroller, digital coding metasurface, and vector network analyzer (VNA). The whole metasurface is designed and fabricated using printed circuit board (PCB) technology, which consists of $3 \times 4$ metasurface panels, and each panel consists of $8 \times 8$ macro meta units. More details about meta unit can be found in Methods and Supplementary Fig. 2. The metasurface is controlled by a Cycone-4 FPGA with the clock frequency of 300 MHz. To reduce the complexity of writing command and controlling system, three SN74LV595A 8-

bit shift registers (https://www.ti.com/lit/ds/symlink/sn74lv595a.pdf?HQS=TI-null-null-alldatasheets-df-pf-SEP-wwe) are utilized to control the rest of PIN diodes sequentially. Moreover, when the PIN diodes are controlled in parallel by a FPGA, three operation cycles are needed after compiling, and thus the total image switching time is around tens of nanoseconds. The SPI (Serial Programming Interface Bus)-controlled command and data between the host computer and FPGA, and VNA are communicated via Ethernet (IEEE 488.2). In addition, the high-level system control code is written in Visual Basic (VB) and Matlab.

To calibrate the measurement system and measure the normalized images, an experimental set up consisting of an anechoic chamber with the size of $2 \times 2 \times 2\ m^3$, including a transmitting (Tx) horn antenna, a waveguide probe receiver, and a vector network analyzer (VNA, Agilent E5071C), was used. In our imaging experiments, VNA is used to acquire the response data by measuring the transmission coefficients ($S_{21}$). To suppress the measurement noise level, the intermediate bandwidth in VNA is set to 10 kHz, respectively. The radio is capable of 70 frequency sweeps per second for 100 points between 3 GHz and 3.4 GHz. As for the data acquisition time, it takes around 0.016 s on average per period. Both the feeding antenna and sample are coaxially mounted on a board at a distance of 1.8 m, since the feeding antenna should be placed in the Fraunhofer region of the coding metasurface. We remark that the aforementioned distance of far-field was obtained from the assumption of point source excitation. In the experiments, however, the incident wave generated from the horn antenna is relatively flat, and thus the corresponding distance may be reduced accordingly.

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

## Acknowledgements

This work was supported in part by the National Key Research and Development Program of China (2017YFA0700201, 2017YFA0700202, and 2017YFA0700203), in part by the National Natural Science Foundation of China (61471006, 61631007, 61571117, and 61731010), in part by the National Science Foundation, and in part by the 111 Project (111–2–05).

## Author contributions

L.L. conceived the idea, conducted the numerical simulations and theoretical analysis. T.J.C. proposed the concept of digital coding and programmable metasurfaces, and L.L., T.J.C., C.W.Q., A.A., and Y.L. wrote the manuscript. All authors participated in the experiments, data analysis, and read the manuscript.

## Additional information

**Competing interests:** The authors declare no competing interests.

