## [Peer Review File · Nature Communications]

Reviewers' Comments:

Reviewer #1:

Remarks to the Author:

Review of « Real-Time Microwave Imager Based on Digital Coding Metasurface »

This paper proposes to develop a microwave computational imaging system by tailoring the radiated patterns according to the type of scene to be imaged. In comparison with existing techniques in the literature based essentially on frequency-diverse random pattern radiation, it is thus proposed to reduce the number of measurements needed to image targets assuming a prior knowledge of their nature. The proposed approach based on the PCA is interesting and innovative, and convincing experimental results are presented in the paper.

However, I note a number of blocking points that do not make it possible for me to recommend the publication of this article in its current state:

INTRODUCTION

The technique is presented as a solution to the technological limitations of existing techniques and prototypes. First, it is argued that current techniques are based on solving an inverse problem for each image, making them ineffective in realistic applications, lines 19-22:

"Even though traditional imagers may be enhanced by compressive sensing and other computational methods, they are typically required to solve the inverse scattering problem again and again when the scene changes, making them largely ineffective for complex in-situ sensing and monitoring."

Although it is understandable that the authors try to highlight the advantages of the proposed technique, this description of the state of the art is nevertheless false. Whether conventional or computational imaging systems can rely on either direct or iterative inversion techniques. In direct inversion techniques where the pseudo-inverse of the propagation operator can be stored in memory, the reconstruction of an image requires as little effort as a matrix multiplication. Moreover, these techniques are not based on any prior knowledge of the nature of the target to be imaged other than their size for the definition of the FOV in opposition to that introduced in this paper. The technique should then only be presented as a solution to reduce the size of these matrices provided that the nature of the scene to be imaged is known.

Line 49-52 state that:

"The data processing is limited by the physical property of the imager, enabling just a few degrees of freedom that can be tuned to optimize the quality. The optimization solves an inverse problem for each individual scene, which is time and resource consuming."

As mentioned previously, it is not necessary to solve the inverse problem for each image calculation in a computational imaging application. In addition, several applications rely on the use of components with a quality factor of a few hundreds or thousands. Thus the expression "just a few degrees of freedom" seems inappropriate to describe these systems.

See for example :

Marks, D. L., Gollub, J., & Smith, D. R. (2016). Spatially resolving antenna arrays using frequency diversity. *JOSA A*, 33(5), 899-912.

Fromenteze, T., Yurduseven, O., Boyarsky, M., Gollub, J., Marks, D. L., & Smith, D. R. (2017). Computational polarimetric microwave imaging. *Optics express*, 25(22), 27488-27505.

The proposed solution is also highlighted by its compatibility with a wide variety of imaging targets in contrast to existing solutions. Lines 52-55 :

"For a data set with large size and variety, it is unlikely that the solution for one scene is also optimized for other scenes. More often, proper configurations for some scenes are inaccessible by the imager, and hence either many different imagers must be used, or one imager is used without consistent efficiency in performance".

In reality, existing solutions based on the use of random radiation patterns do not impose any restrictions on the nature of the scene to be imaged other than those related to the size and sampling of the radiating aperture. This property is guaranteed by the random nature of the H matrix linking the target domain to the measured modes.

Baraniuk, Richard, et al. "A simple proof of the restricted isometry property for random matrices." *Constructive Approximation* 28.3 (2008): 253-263.

On the other hand, the proposed technique must face such limitations insofar as the set of modes interrogating the scene must be specifically chosen for this type of scene. It is therefore necessary to clarify the advantage of the proposed technique, allowing only the number of measurements to be limited compared to a purely random system.

RESULTS

It seems absolutely necessary to introduce a schematic representation or a photograph of the complete system in order to facilitate the reproducibility of these results. It is difficult to have a good representation of the 16 metasurfaces based only on the schematic view of the unit cell described in this section.

Considering the remarks made in the previous section, I do not think that Figure 1a correctly represents the difference between the existing systems and the proposed approach. What is referred to as "fixed imager 1" and "fixed imager 2" occupies a limited space from space to M dimensions compared to spaces occupied by "digital metasurfaces". If these systems are based on random radiation patterns, they should occupy a larger space than those corresponding to the application presented if comparable radiating aperture size and spatial sampling are considered and as long as the quality factor of the components is high.

One of the advantages put forward for this technique is the ability to reconstruct images in real time by reducing the number of modes required with the use of PCA. However, important data concerning the performance of the computing unit used and the type of communication chosen between the VNA and this machine are also necessary for the reproducibility of these results. It is also desirable to mention the average acquisition and computation times according to the characteristics of the computing unit in use. It would also be interesting to give the training time required to extract the principal components, for example in the case of a moving person.

Line 198: "Note that both the head and feet cannot be clearly identified, due to the limited size of the metasurface with respect to the one of the person under consideration."

The size of the metasurface should primarily define the resolution of the subsystem and not the FOV (unless directive sources are used). A limited FOV may be a sign of secondary sources that are too directive or separated by more than half the operating wavelength.

The results obtained are compared with those of random metasurfaces. Please explicitly specify the reconstruction technique considered as it has a major impact on the quality of the retrieved

images. Throughout the document, it seems that the reconstructed images are never supported by a colorbar giving valuable information about the dynamics of the system. It seems necessary to at least indicate if the images are represented according to a linear or logarithmic scale, if a thresholding is done or any other image processing.

METHODS

Lines 338-339 : "Although the random projection is near-isometric, it is data oblivious and hence cannot leverage any special geometric structure of the dataset. PCA is a data-aware linear embedding technique, where each row of H (i.e., the measurement mode) is trained over many training samples available. "

This section allows, in my opinion, to better highlight the interest of the proposed technique in comparison with the explanations given previously. It seems that a random approach imposes no restrictions on the nature of the object to be reconstructed but that an approach based on PCA reduces the number of required measurements by using a prior knowledge on its nature. Thus, the illustrations previously proposed to compare random systems with trained systems do not seem suitable to me.

Minor typos:

Line 172: "measurement mods"

Supplementary Note 1: The 3D free space Green's function is usually written with a minus sign on the phase to ensure causality. The impact here is minor since the presented technique is operating at a single frequency.

Synthesis of comments:

In my opinion, the technique proposed in this article is of particular interest. The existing systems in the literature are based on the exploitation of structures inherent to the scenes to be imaged. By using a prior knowledge of their nature, it is proposed to tailor the radiation in a set of modes whose linear combinations allow relatively accurate representations of these scenes. For a given configuration, the diversity of targets that can be imaged is thus limited but the number of measurements required is also reduced, which represents a substantial advance in the field of computational imaging. In addition, this proof of concept validates the possible implementation of this technique in real-time applications.

I think, however, that the paper needs to be reworked to present a more realistic description of the state of the art. The description of systems based on random radiation seems to me to be false in that there are no restrictions imposed on the nature of the imaged scenes.

In my opinion, paper also suffers from an uncomplete description of the system and the methods developed. It is essential that the document contain sufficient information to allow the results to be reproduced.

Reviewer #2:

Remarks to the Author:

The authors present a reprogrammable imager on a 2-bit reprogrammable coding metasurface, which is trained with machine-learning techniques at the physical level. They use the system to monitor and recognize human movement in real-time, and also test the system on the MNIST dataset of handwritten digits. I believe that the manuscript presents an incremental advance over

what has been shown in the literature, and therefore I cannot recommend it for publication due to lack of novelty.

Nature Communications suggests that referees consider a few points for review including, "On a more subjective note, do you feel that the paper will influence thinking in the field?" In all honesty I have to respond that, no I do not feel this article will influence thinking in the field, since researchers in the field have already published on many aspects presented in the paper, such as: dynamic metasurfaces, imaging with static and dynamic metasurfaces, digital and analog encodings with metasurfaces, machine learning with metasurfaces, and microwave metasurfaces for imaging. (I do not provide references for this prior work I mention above, since all of these works are cited as references in the paper.) What the authors have done is to make a straightforward, but incremental, advance over prior work, some of which is their own, and thus the work lacks novelty.

Reviewer #3:

Remarks to the Author:

Real Time Microwave Imager Based on Digital Coding Metasurface is an interesting article with results that potentially justify publication in Nature Communications. However, it is hard to fully understand the methods and results in the current context. More detail with regard to the system parameters and model and the processing methods should be included in the text and supplemental material.

For example, the scene is 3D but modeled as 2D. The PCA refers to PCA of the forward model? Doesn't the projection of the forward model on a 2D manifold affect this? its not very easy to understand. Also the scene should be complex, but the PCA is on a real system. How was phase managed. Why is there little or no speckle in the imaging system? I realize that these might seem clear or trivial issues to the authors, but their answers are not immediately clear to this reviewer.

Along similar lines, more detailed system performance metrics with regard to expected and achieved resolution and compression ratios are needed. One number, 7.5% data usage relative to full sampling, is provided, but it is not clear how this is calculated. A typical scientific study in this space contains more detailed discussion of SNR, resolution and compression metrics.

Finally, other than the fact that a neural network was used for data processing remarkably little information is provided as to the structure and performance of this network is provided. It would be impossible for an expert to replicate this work with the level of detail present here, in a scientific study detailed description of the network, the training format etc. should be provided.

The metamaterial structures themselves are interesting and the use of machine learning to implement this style of imaging is definitely the right direction to go.

Responses to Reviewers' Comments

We thank all reviewers for their constructive and valuable comments, which helped improve the quality of the manuscript significantly. The replies to all comments are provided below in blue colors, and all changes in the revised manuscript have been highlighted in yellow background.

To Reviewer #1:

Comment:

This paper proposes to develop a microwave computational imaging system by tailoring the radiated patterns according to the type of scene to be imaged. In comparison with existing techniques in the literature based essentially on frequency-diverse random pattern radiation, it is thus proposed to reduce the number of measurements needed to image targets assuming a prior knowledge of their nature. The proposed approach based on the PCA is interesting and innovative, and convincing experimental results are presented in the paper.

Response:

Thank you very much for your kind support and encouraging comments.

Comment:

Q1. The technique is presented as a solution to the technological limitations of existing techniques and prototypes. First, it is argued that current techniques are based on solving an inverse problem for each image, making them ineffective in realistic applications, lines 19-22:

"Even though traditional imagers may be enhanced by compressive sensing and other computational methods, they are typically required to solve the inverse scattering problem again and again when the scene changes, making them largely ineffective for complex in-situ sensing and monitoring."

Although it is understandable that the authors try to highlight the advantages of the proposed technique, this description of the state of the art is nevertheless false. Whether conventional or computational imaging systems can rely on either direct or iterative inversion techniques. In direct inversion techniques where the pseudo-inverse

of the propagation operator can be stored in memory, the reconstruction of an image requires as little effort as a matrix multiplication. Moreover, these techniques are not based on any prior knowledge of the nature of the target to be imaged other than their size for the definition of the FOV in opposition to that introduced in this paper. The technique should then only be presented as a solution to reduce the size of these matrices provided that the nature of the scene to be imaged is known.

Line 49-52 state that:

"The data processing is limited by the physical property of the imager, enabling just a few degrees of freedom that can be tuned to optimize the quality. The optimization solves an inverse problem for each individual scene, which is time and resource consuming."

As mentioned previously, it is not necessary to solve the inverse problem for each image calculation in a computational imaging application. In addition, several applications rely on the use of components with a quality factor of a few hundreds or thousands. Thus the expression "just a few degrees of freedom" seems inappropriate to describe these systems.

See for example :

Marks, D. L., Gollub, J., & Smith, D. R. (2016). Spatially resolving antenna arrays using frequency diversity. *JOSA A*, 33(5), 899-912.

Fromenteze, T., Yurduseven, O., Boyarsky, M., Gollub, J., Marks, D. L., & Smith, D. R. (2017). Computational polarimetric microwave imaging. *Optics express*, 25(22), 27488-27505.

The proposed solution is also highlighted by its compatibility with a wide variety of imaging targets in contrast to existing solutions. Lines 52-55 :

"For a data set with large size and variety, it is unlikely that the solution for one scene is also optimized for other scenes. More often, proper configurations for some scenes are inaccessible by the imager, and hence either many different imagers must be used, or one imager is used without consistent efficiency in performance".

In reality, existing solutions based on the use of random radiation patterns do not impose any restrictions on the nature of the scene to be imaged other than those related to the size and sampling of the radiating aperture. This property is guaranteed by the random nature of the H matrix linking the target domain to the measured

modes.

On the other hand, the proposed technique must face such limitations insofar as the set of modes interrogating the scene must be specifically chosen for this type of scene. It is therefore necessary to clarify the advantage of the proposed technique, allowing only the number of measurements to be limited compared to a purely random system

Response:

Thank you very much for these insightful comments and suggestions. The reviewer pointed out two unsuitable descriptions mentioned above, and suggested that it is more suitable to state that the proposed method provides a solution to reduce remarkably the size of these matrices provided that the nature of the scene to be imaged is known.

Indeed, some existing imaging techniques do not rely on any knowledge about the scene (except that the compressive sensing based imager utilize the sparsity of the scene), but they require large sizes of dense matrices, and hence they are limited in the efficiency or accuracy. Incorporating knowledge about the nature of scene is thus beneficial and commonly used in many computational imaging techniques. We mainly address such types of imaging approaches.

According to the suggestions by the reviewer, we have revised the statements as follows:

(1) Conventional imagers usually require either time-consuming data acquisition, or complicated reconstruction algorithms for data post-processing, making them largely ineffective for complex in-situ sensing and monitoring.

(2) Recently, compressive sensing inspired computational imagers have been proposed to reduce remarkably hardware cost and speed of data acquisition, which are at the cost of iterative reconstruction algorithms. The optimization solves a time and resource consuming inverse problem for each individual scene. In this sense, it is typically required to solve the inverse scattering problem again and again when the scene changes, making them largely ineffective for complex in-situ sensing and monitoring.

(3) As the reviewer mentioned, the reconstruction with random radiation patterns is certainly possible, which is independent of the scene. However, we remark that the quality of the reconstructed image with random patterns is comparably lower when

the same number of measurements is made (as shown in Fig. 3). It is the independence of scene that makes the random approach un-optimized. We agree that our previous statement is inaccurate: *the optimal quality is accessible when the measurement number keeps increasing*. We have removed this description in the revised manuscript.

Finally, we appreciate the reviewer for bringing these relevant references to our attention. They have been cited properly.

Comment:

Q2: It seems absolutely necessary to introduce a schematic representation or a photograph of the complete system in order to facilitate the reproducibility of these results.

Response:

Thank you very much for your good suggestion. A representation of the complete system along with detailed description have been added in Supplementary Figure S1 and Measurement System of Methods, respectively.

Comment:

Q3: Considering the remarks made in the previous section, I do not think that Figure 1a correctly represents the difference between the existing systems and the proposed approach. What is referred to as "fixed imager 1" and "fixed imager 2" occupies a limited space from space to M dimensions compared to spaces occupied by "digital metasurfaces". If these systems are based on random radiation patterns, they should occupy a larger space than those corresponding to the application presented if comparable radiating aperture size and spatial sampling are considered and as long as the quality factor of the components is high.

Response:

Thank you very much for your valuable comments. Fig. 1a represents the space that a fixed number of measurements can generate, but not the space to be occupied for imaging the object. The reviewer is certainly correct in the sense that the fixed imager can occupy larger configuration space with larger aperture size, spatial sampling, and quality factor. Nevertheless, these factors will influence the digital metasurface in the same way. Hence there is no contradiction and the digital metasurface still occupies a huge configuration space. For a fixed metasurface, an incident plane wave at a fixed frequency can only generate one radiation pattern, while our two-bit digital

metasurface can generate 4^N patterns with N being the number of meta-atoms. That marks our key distinction. For frequency-sweeping computational imagers, because as the reviewer said, the patterns should occupy a larger space than that represents the object. Our digital metasurface makes it possible to work well even at just a single frequency, and more importantly, our metasurface imager could be reprogrammable and trained to accommodate the change of object. In principle, if more frequencies are used in our metasurface imager, its configuration space will further grow as the previous imager.

Therefore, we really wish to maintain Fig. 1a since it presents the most significant distinction of our concept. We have put some extra texts to elaborate this in the revised manuscript.

Comment:

Q4: One of the advantages put forward for this technique is the ability to reconstruct images in real time by reducing the number of modes required with the use of PCA. However, important data concerning the performance of the computing unit used and the type of communication chosen between the VNA and this machine are also necessary for the reproducibility of these results. It is also desirable to mention the average acquisition and computation times according to the characteristics of the computing unit in use. It would also be interesting to give the training time required to extract the principal components, for example in the case of a moving person.

Response:

Thank you very much for your constructive suggestions. In the revised manuscript, we have detailed these important data in **Discussions** and **Methods**, which are highlighted in *yellow* background.

Comment:

Q5: Line 198: "Note that both the head and feet cannot be clearly identified, due to the limited size of the metasurface with respect to the one of the person under consideration."

The size of the metasurface should primarily define the resolution of the subsystem and not the FOV (unless directive sources are used). A limited FOV may be a sign of secondary sources that are too directive or separated by more than half the operating wavelength.

Response:

Thank you very much for the valuable comments. It is indeed the case that we used a directive receiver (horn antenna), which leads to the limited FOV. We have revised the discussions in the revised manuscript.

Comment:

Q6: The results obtained are compared with those of random metasurfaces. Please explicitly specify the reconstruction technique considered as it has a major impact on the quality of the retrieved images. Throughout the document, it seems that the reconstructed images are never supported by a colorbar giving valuable information about the dynamics of the system. It seems necessary to at least indicate if the images are represented according to a linear or logarithmic scale, if a thresholding is done or any other image processing.

Response:

Sorry for these confusions, and thank you very much for pointing out these important issues. Regarding to the reconstruction algorithm, we have used the pseudo-inverse operation for imaging, as described in *Imaging Modelling* of *Methods*. Additionally, we used the k-cluster algorithm for the object recognition, as described in the second paragraph of *Object Recognition in Compressed Domain*. Regarding to the reconstruction results, we have added the color bar in the revised manuscript.

Comment:

Q7: Lines 338-339 : "Although the random projection is near-isometric, it is data oblivious and hence cannot leverage any special geometric structure of the dataset. PCA is a data-aware linear embedding technique, where each row of H (i.e., the measurement mode) is trained over many training samples available. " This section allows, in my opinion, to better highlight the interest of the proposed technique in comparison with the explanations given previously.

Response:

Thank you very much for this very insightful suggestion. We have revised the above description as follows:

The random approach imposes no restrictions on the nature of the object to be reconstructed. On the contrary, PCA is a prior data aware linear embedding technique, where each row of H (i.e., the measurement mode) is trained over many training samples available. In this way, when a set of prior knowledge on the scene

under investigation is available, the PCA approach enables the design of efficient measurement matrices, allowing the number of measurements to be much smaller than that in the purely random system.

Comment:

Q8: Minor issues: Line 172: "measurement mods" Supplementary Note 1: The 3D free space Green's function is usually written with a minus sign on the phase to ensure causality. The impact here is minor since the presented technique is operating at a single frequency.

Response:

Sorry for this confusion, and thank you very much to point out this issue. It has been fixed in the revised manuscript.

Comment:

In my opinion, the technique proposed in this article is of particular interest. The existing systems in the literature are based on the exploitation of structures inherent to the scenes to be imaged. By using a prior knowledge of their nature, it is proposed to tailor the radiation in a set of modes whose linear combinations allow relatively accurate representations of these scenes. For a given configuration, the diversity of targets that can be imaged is thus limited but the number of measurements required is also reduced, which represents a substantial advance in the field of computational imaging. In addition, this proof of concept validates the possible implementation of this technique in real-time applications.

I think, however, that the paper needs to be reworked to present a more realistic description of the state of the art. The description of systems based on random radiation seems to me to be false in that there are no restrictions imposed on the nature of the imaged scenes.

In my opinion, paper also suffers from an uncomplete description of the system and the methods developed. It is essential that the document contain sufficient information to allow the results to be reproduced.

Response:

Thank you very much for your supportive and constructive comments and suggestions. We have revised our manuscript based on these suggestions, and added the necessary materials to explain our method.

To Reviewer #2:

Comment:

The authors present a reprogrammable imager on a 2-bit reprogrammable coding metasurface, which is trained with machine-learning techniques at the physical level. They use the system to monitor and recognize human movement in real-time, and also test the system on the MNIST dataset of handwritten digits. I believe that the manuscript presents an incremental advance over what has been shown in the literature, and therefore I cannot recommend it for publication due to lack of novelty.

Nature Communications suggests that referees consider a few points for review including, which is trained with machine-learning techniques at the physical level. They use the system to monitor and recognize human movement in real-time, and also test the system on the MNIST dataset of handwritten digits. In all honesty I have to respond that, no I do not feel this article will influence thinking in the field, since researchers in the field have already published on many aspects presented in the paper, such as: dynamic metasurfaces, imaging with static and dynamic metasurfaces, digital and analog encodings with metasurfaces, machine learning with metasurfaces, and microwave metasurfaces for imaging. (I do not provide references for this prior work I mention above, since all of these works are cited as references in the paper.) What the authors have done is to make a straightforward, but incremental, advance over prior work, some of which is their own, and thus the work lacks novelty.

Response:

With all due respect, we feel that the reviewer's comments are quite inaccurate. The reviewer said that the references cited in our manuscript have largely done what we have done. That is not true at all. We understand that if one simply watches upon the keywords of "dynamic metasurface", "machine learning with metasurfaces", "digital metasurface", and so on, of course it is not novel now as the reviewer said. However, we cannot just superficially judge the novelty by the keywords.

In fact, the proposed method in this manuscript exactly influences the thinking in this field and the interdisciplinary fields. We have made very clear benchmarking statements against the state-of-the-arts, and the digital metasurface and machine learning algorithm are indeed not our main contributions in this work. Instead, we have synergized those two building blocks, and ignited new trends in metasurfaces which could be trained for in-situ and reprogrammable imaging. That is drastically distinct from the traditional metasurface imaging. In addition to the summary stated

clearly in the manuscript, please kindly refer to our reply to Q3 of Reviewer #1.

To Reviewer #3:

Comment:

Real Time Microwave Imager Based on Digital Coding Metasurface is an interesting article with results that potentially justify publication in Nature Communications. However, it is hard to fully understand the methods and results in the current context. More detail with regard to the system parameters and model and the processing methods should be included in the text and supplemental material.

Response:

Thank you very much for your kind support and encouraging comments.

Comment:

Q1. For example, the scene is 3D but modeled as 2D. The PCA refers to PCA of the forward model? Doesn't the projection of the forward model on a 2D manifold affect this? its not very easy to understand. Also the scene should be complex, but the PCA is on a real system. How was phase managed. Why is there little or no speckle in the imaging system?

Response:

Thank you very much for raising this important question. It is true that we modeled 3D scene as its 2D projection, and we used the PCA machine learning technique to learn the relation between the measurements and the 2D scene projection.

It is true that the theoretical PCA modes are real-valued. However, the corresponding physical PCA modes generated by metasurface using the G-S algorithm are complex-valued. For the sake of plotting, we just provided their amplitudes.

Regarding speckles, there are three possible reasons. First, the reconstruction algorithm is the pseudo-inverse operation, where the smoothing solution is implicitly imposed. Second, we use the PCA machine learning technique, and the leading PCA modes play the role of low-pass filters. Finally, within an image resolution, the scene structure is relatively simple, where the complicated multiple scattering effects do not exit, which is possibly responsible for no speckle in images.

Comment:

Q2. Along similar lines, more detailed system performance metrics with regard to expected and achieved resolution and compression ratios are needed. One number, 7.5% data usage relative to full sampling, is provided, but it is not clear how this is calculated. A typical scientific study in this space contains more detailed discussion of SNR, resolution and compression metrics.

Response:

Thank you very much for these constructive comments and suggestions. Regarding to the compression ratio, it is defined as the ratio of the number of measurements to the number of numerical image pixels. Further, the value of 7.5% comes from 60 measurements and 800 image pixels.

We investigated the performance of the proposed method via a selected set of numerical simulations, as shown in Supplementary Figure S3, where the number of measurements, and SNR have been analyzed.

Comment:

Q3. Finally, other than the fact that a neural network was used for data processing remarkably little information is provided as to the structure and performance of this network is provided. It would be impossible for an expert to replicate this work with the level of detail present here, in a scientific study detailed description of the network, the training format etc. should be provided.

Response:

Sorry for this confusion, and thank you very much for your suggestion. More details about the data training have been added in *Supplementary Note 2*.

Comment:

The metamaterial structures themselves are interesting and the use of machine learning to implement this style of imaging is definitely the right direction to go.

Response:

Thank you very much for your kind supports and encouraging comments.

Reviewers' Comments:

Reviewer #1:

Remarks to the Author:

All my remarks seem to have been taken into consideration and the paper is now, in my opinion, sufficiently complete to recommend its publication, given the significant progress it offers on this new field.

Some remarks:

1. I realized when I reread my remarks that the FOV is probably also limited by the size of the metasurface, as initially suggested in the initial version of the paper, although it seems thanks to the modifications made to the paper that the horn used can also have an impact.

The FOV must be directly related to the sampling of the secondary sources constituting the metasurface in the case of point object imaging. However, the maximum size of images made with continuous objects will indeed be directly affected by the size of the radiating aperture due to their specular nature.

2. An internet search of the shift register "SN74ALV595A", which was unknown to me, did not lead to any results. It is likely that there is an error in the name of the reference.

Reviewer #3:

Remarks to the Author:

I thank the authors for their revised manuscript, which substantially clarifies what they are doing. Unfortunately, this further clarity leads me to doubt the novelty and significance of the manuscript. Previously I thought that the authors were using artificial intelligence in the form of neural networks to match the measurement modes and to complete the inverse operation. This approach is described, for example in Liu, Z., Zhu, D., Rodrigues, S. P., Lee, K. T., & Cai, W. (2018). Generative Model for the Inverse Design of Metasurfaces. *Nano letters*, 18(10), 6570-6576.

With the further clarity of supplementary note 2, however, it seems that the authors are simply using the G-S algorithm to find the antenna pattern that produces the desired field. Since the G-S algorithm has been used for this kind of thing for around 40 years, its use here hardly qualifies as "machine learning". In this sense, I feel that the claims made in the article to be using machine learning are misleading and that the article is less novel than I originally thought. I also feel that in this regard the design strategy for the illumination modes and, most particularly, the linear inverse approach adopted are not competitive with the current state of the art in metasurface imaging.

Responses to the Reviewers' Comments

Reviewer #1:

Comment:

All my remarks seem to have been taken into consideration and the paper is now, in my opinion, sufficiently complete to recommend its publication, given the significant progress it offers on this new field.

Response:

Thank you for your insightful comments and kind supports!

Some remarks:

1. I realized when I reread my remarks that the FOV is probably also limited by the size of the metasurface, as initially suggested in the initial version of the paper, although it seems thanks to the modifications made to the paper that the horn used can also have an impact.

The FOV must be directly related to the sampling of the secondary sources constituting the metasurface in the case of point object imaging. However, the maximum size of images made with continuous objects will indeed be directly affected by the size of the radiating aperture due to their specular nature.

Response:

Thank you very much for your insightful comments. We have accordingly revised our manuscript, as following:

Note that both the head and feet cannot be clearly identified due to the limited FOV, which probably arises from two reasons, i.e., the use of a directive receiver (horn antenna), and the limited size of the metasurface with respect to the person under consideration.

2. An internet search of the shift register "SN74ALV595A", which was unknown to me, did not lead to any results. It is likely that there is an error in the name of the reference.

Response:

Sorry for this typo. The name of the shift register has been corrected as SN74LV595A in this revised version.

Reviewer #3:

Comment:

I thank the authors for their revised manuscript, which substantially clarifies what they are doing. Unfortunately, this further clarity leads me to doubt the novelty and significance of the manuscript. Previously I thought that the authors were using artificial intelligence in the form of neural networks to match the measurement modes and to complete the inverse operation. This approach is described, for example in Liu, Z., Zhu, D., Rodrigues, S. P., Lee, K. T., & Cai, W. (2018). Generative Model for the Inverse Design of Metasurfaces. *Nano letters*, 18(10), 6570-6576.

With the further clarity of supplementary note 2, however, it seems that the authors are simply using the G-S algorithm to find the antenna pattern that produces the desired field. Since the G-S algorithm has been used for this kind of thing for around 40 years, its use here hardly qualifies as "machine learning" In this sense, I feel that the claims made in the article to be using machine learning are misleading and that the article is less novel than I originally thought. I also feel that in this regard the design strategy for the illumination modes and, most particularly, the linear inverse approach adopted are not competitive with the current state of the art in metasurface imaging.

Response:

Sorry for this confusion.

It seems that the reviewer has misunderstood our approach, which is certainly not just the G-S algorithm. In fact, it is impossible to find the antenna pattern (the configurations of the reprogrammable coding metasurface) real-timely as we did by only using the G-S algorithm. **A machine learning method** is needed to obtain the desirable radiation patterns (Step 1). Then the G-S algorithm is simply used to infer the configurations of reprogrammable metasurface from the obtained radiation patterns (Step 2). Thereby the real-time high-quality imaging and high-accuracy object recognition can be achieved by using the significantly reduced measurements.

In Step 1, we train our reprogrammable imager with PCA, **a standard machine learning technique**, which justifies our usage of the term machine learning. We did

not use the neural networks because we are more focused on the physical implementation over a reprogrammable coding metasurface, not joining the competition arena of the algorithm development. In our work, PCA is adopted as an illustrative tool to realize the powerful reprogrammable microwave imager, which can be trained to drastically enhance the imaging speed, performance, and scene adaptability thanks to the machine learning algorithm and the reprogrammable coding metasurface.

In Step 2, the G-S algorithm is certainly a long-used method, but *it is used to facilitate the results of Step 1 and serves as a pavement, rather than our major claim*. The G-S algorithm can be easily replaced by other methods, but any choice in Step 2 does not harm the novelty of our work.

Reviewers' Comments:

Reviewer #1:

Remarks to the Author:

Nothing more to add for my part, I recommend the publication of the article as it stands.

Reviewer #3:

Remarks to the Author:

I thank the authors for their response to my concerns regarding their paper "Machine-Learning Reprogrammable Metasurface Imager." Unfortunately, with each step of explanation my concern that the title and presentation of this paper is misleading grows. In current usage, e.g. papers published in the past couple of years, machine learning almost universally refers to deep learning. While, as the authors indicate in their response, principal component analysis is a "machine learning technique," algebra is also a machine learning technique, but algebra is not machine learning.

The authors indicate in their latest response that "We did not use the neural networks because we are more focused on the physical implementation over a reprogrammable coding metasurface, not joining the competition arena of the algorithm development." This is a fair statement, but in view of this statement the title of the paper and the presentation should focus on the novelty of this physical component, not on the claim that the paper uses machine learning to reprogram a metasurface. PCA is a linear algebra technique, it is even less central to modern machine learning than the GS algorithm. While it is fair to say that the authors do not want to join the competitive area of machine learning algorithm development, in this case they should not present the paper as though it were a machine learning paper.

Responses to the Reviewers' Comments

Reviewer #1:

Comment:

All my remarks seem to have been taken into consideration and the paper is now, in my opinion, sufficiently complete to recommend its publication, given the significant progress it offers on this new field.

Response:

Thank you for your insightful comments and kind supports!

Some remarks:

3. I realized when I reread my remarks that the FOV is probably also limited by the size of the metasurface, as initially suggested in the initial version of the paper, although it seems thanks to the modifications made to the paper that the horn used can also have an impact.

The FOV must be directly related to the sampling of the secondary sources constituting the metasurface in the case of point object imaging. However, the maximum size of images made with continuous objects will indeed be directly affected by the size of the radiating aperture due to their specular nature.

Response:

Thank you very much for your insightful comments. We have accordingly revised our manuscript, as following:

Note that both the head and feet cannot be clearly identified due to the limited FOV, which probably arises from two reasons, i.e., the use of a directive receiver (horn antenna), and the limited size of the metasurface with respect to the person under consideration.

4. An internet search of the shift register "SN74ALV595A", which was unknown to me, did not lead to any results. It is likely that there is an error in the name of the reference.

Response:

Sorry for this typo. The name of the shift register has been corrected as SN74LV595A

in this revised version.

Reviewer #3:

Comment:

I thank the authors for their revised manuscript, which substantially clarifies what they are doing. Unfortunately, this further clarity leads me to doubt the novelty and significance of the manuscript. Previously I thought that the authors were using artificial intelligence in the form of neural networks to match the measurement modes and to complete the inverse operation. This approach is described, for example in Liu, Z., Zhu, D., Rodrigues, S. P., Lee, K. T., & Cai, W. (2018). Generative Model for the Inverse Design of Metasurfaces. *Nano letters*, 18(10), 6570-6576.

With the further clarity of supplementary note 2, however, it seems that the authors are simply using the G-S algorithm to find the antenna pattern that produces the desired field. Since the G-S algorithm has been used for this kind of thing for around 40 years, its use here hardly qualifies as "machine learning" In this sense, I feel that the claims made in the article to be using machine learning are misleading and that the article is less novel than I originally thought. I also feel that in this regard the design strategy for the illumination modes and, most particularly, the linear inverse approach adopted are not competitive with the current state of the art in metasurface imaging.

Response:

Sorry for this confusion.

It seems that the reviewer has misunderstood our approach, which is certainly not just the G-S algorithm. In fact, it is impossible to find the antenna pattern (the configurations of the reprogrammable coding metasurface) real-timely as we did by only using the G-S algorithm. **A machine learning method** is needed to obtain the desirable radiation patterns (Step 1). Then the G-S algorithm is simply used to infer the configurations of reprogrammable metasurface from the obtained radiation patterns (Step 2). Thereby the real-time high-quality imaging and high-accuracy object recognition can be achieved by using the significantly reduced measurements.

In Step 1, we train our reprogrammable imager with PCA, **a standard machine learning technique**, which justifies our usage of the term machine learning. We did not use the neural networks because we are more focused on the physical

implementation over a reprogrammable coding metasurface, not joining the competition arena of the algorithm development. In our work, PCA is adopted as an illustrative tool to realize the powerful reprogrammable microwave imager, which can be trained to drastically enhance the imaging speed, performance, and scene adaptability thanks to the machine learning algorithm and the reprogrammable coding metasurface.

In Step 2, the G-S algorithm is certainly a long-used method, but *it is used to facilitate the results of Step 1 and serves as a pavement, rather than our major claim*. The G-S algorithm can be easily replaced by other methods, but any choice in Step 2 does not harm the novelty of our work.